# Similar Theory of Mind Deficits in Community Dwelling Older Adults with Vascular Risk Profile and Patients with Mild Cognitive Impairment: The Case of Paradoxical Sarcasm Comprehension

**DOI:** 10.3390/brainsci11050627

**Published:** 2021-05-13

**Authors:** Glykeria Tsentidou, Despina Moraitou, Magda Tsolaki

**Affiliations:** 1Laboratoty of Psychology, Department of Experimental and Cognitive Psychology, School of Psychology, Aristotle University of Thessaloniki, 54124 Thessaloniki, Greece; despinamorait@gmail.com; 2Laboratory of Neurodegenerative Diseases, Center for Interdisciplinary Research and Innovation (CIRI), Aristotle University of Thessaloniki, 57001 Thessaloniki, Greece; tsolakim1@gmail.com; 3Health Center of Katerini, 60100 Pieria, Greece; 4Greek Association of Alzheimer’s Disease and Related Disorders (GAADRD), 54643 Thessaloniki, Greece; 51st Department of Neurology, Medical School, Aristotle University of Thessaloniki, 54124 Thessaloniki, Greece

**Keywords:** vascular pathology, ToM, MCI, sarcasm comprehension, Alzheimer’s disease dementia

## Abstract

Recent studies deal with disorders and deficits caused by vascular syndrome in efforts for prediction and prevention. Cardiovascular health declines with age due to vascular risk factors, and this leads to an increasing risk of cognitive decline. Mild cognitive impairment (MCI) is defined as the negative cognitive changes beyond what is expected in normal aging. The purpose of the study was to compare older adults with vascular risk factors (VRF), MCI patients, and healthy controls (HC) in social cognition and especially in theory of mind ability (ToM). The sample comprised a total of 109 adults, aged 50 to 85 years (M = 66.09, SD = 9.02). They were divided into three groups: (a) older adults with VRF, (b) MCI patients, and (c) healthy controls (HC). VRF and MCI did not differ significantly in age, educational level or gender as was the case with HC. Specifically, for assessing ToM, a social inference test was used, which was designed to measure sarcasm comprehension. Results showed that the performance of the VRF group and MCI patients is not differentiated, while HC performed higher compared to the other two groups. The findings may imply that the development of a vascular disorder affecting vessels of the brain is associated from its “first steps” to ToM decline, at least regarding specific aspects of it, such as paradoxical sarcasm understanding.

## 1. Introduction

Social cognition refers to the ability to identify, observe and interpret socially relevant information. It involves a broad range of abilities that extend to, among others, recognizing emotions, empathy, social knowledge and reasoning, and the theory of mind (ToM). ToM is a broad cognitive process, which includes several sub-processes, such as mental state definition and intention attribution, and more specifically, it encompasses the ability to comprehend and predict the aims, beliefs, desires, thoughts, and behaviors of other individuals, and is considered crucial for representing others’ mental states. ToM is divided into two subcomponents: cognitive and affective ToM [1]. The cognitive subcomponent describes the ability to understand cognitive states (beliefs, thoughts, intentions), while the affective subcomponent describes the ability to perceive emotions (affective states, feelings of others) [2]. ToM subcomponents are different; each involves specific and differentiated neuronal systems. Shamay-Tsoory et al. (2006) [3] displayed that impairments in ToM affective task performance are related to ventromedial prefrontal cortex damage in adults. Kalbe et al. (2010) [4] explored the neural substrates of the ToM cognitive subcomponent; using repetitive transcranial magnetic stimulation, they showed the selective role of dorsolateral prefrontal cortex in ToM cognitive task (but not affective task) performance. The cognitive ToM subcomponent is mostly explored using the false-belief paradigm, through which the ability to comprehend that another person’s beliefs are different from one’s own beliefs and from reality as well is measured, while affective ToM abilities are mostly related to tasks that require determining emotional mental states from images or videos of the entire face area or just the eye region.

### 1.1. Theory of Mind in Typical Aging

Converging evidence suggests that ToM abilities decline with age. However, given that ToM abilities depend on at least ToM-specific processes, it is unclear what leads to the age-related deterioration of ToM abilities. The meta-analysis of Henry et al. (2013) [5] does not answer whether changes in underlying ToM competence, executive functions, or both cause age-related decline of ToM. Most studies have focused on ToM, considered the central process of social cognition, as it is connected to different abilities, such as social-perception processes, emotional processes, empathy and social awareness [6]. Such abilities may also be related to executive function and memory [7]. Kemp et al. (2012), [1] support that age-related decline in social understanding may be partially mediated by declining fluid intelligence. On the other hand, Joseph Moran (2013) [8], assumes that ToM abilities may be relatively spared, suggesting the acquisition of a sort of social wisdom with advancing age. More recently, Hughes et al. (2019) [9] underline that older adults’ performance is poorer in ToM compared to younger adults’ and based on fMRI indications, older adults display weaker intrinsic connectivity between the right temporoparietal junction and right temporal pole, which can explain their lower ToM performance. Charles and Carstensen (2010) [10] aptly report that the cognitive changes that emerge with age paint a complex picture of loss and gain, as some losses in cognitive functioning interact with some gains in knowledge or ability in other domains, which makes it difficult to get a clear picture about theory of mind abilities during the lifespan.

### 1.2. ToM Abilities and Pathological Cognitive Decline

Social cognition disturbances appear in a great deal of neurological and psychiatric disorders and sometimes constitute part of the diagnostic criteria. According to the American Psychiatric Association’s Diagnostic and Statistical Manual for Mental Disorders, (DSM-5), social cognition is considered a vital component of neurocognitive functioning. In the DSM-5, social cognition is distinguished between three subdomains: recognizing emotions, ToM, and insight, which refer to the conscious awareness of the changes related to the disease. For most disorders that show deficits in ToM (frontotemporal lobar degeneration, Alzheimer’s disease, Lewy body disease, Parkinson’s, etc.) the DSM-5 clarifies specific criteria for social cognition disorder [11]. As already mentioned, social cognition disorders appear in a wide range of pathologies. The most recent research work reported that, regarding frontotemporal dementia, social cognition deficit is the key feature of it [12]. In terms of Alzheimer’s disease dementia (ADD), the most common form of dementia, opinions greatly differ about how ADD impacts ToM functioning. A few studies have suggested that ToM deficits display the deterioration of other cognitive functions in individuals with ADD [13,14] and found ToM deficits in ADD that could not be explained by general effects of dementia on executive dysfunction, disrupting the prevailing view that such deficits are secondary to general cognitive processing demands. No correlations were found between significant deficits in ToM and severity of cognitive impairment and executive function, and the deficits in visual perspective taking (transfer inference) were not proven to be caused by primary deficits in visuospatial ability and memory in the AD patients. However, Kamp et al. [1] and Klein Koerkamp et al. [15] argue that ADD patients display various difficulties in processing emotions, ToM, and empathy; disturbances are thought to be caused by a general cognitive impairment impacting their understanding of cognitive tasks [16].

Other studies of ADD in general support that prominent atrophy and tau deposition is apparent in the hippocampus and amygdala and this possibly results in deficits in social cognition [17,18,19]. These neuropathological features are even obvious in earlier prodromal phases, specifically in stages with mild cognitive impairment (MCI) [20]. According to the Alzheimer’s Association, the MCI group includes persons diagnosed with a minor but traceable and measurable decline in cognitive abilities, including memory and thinking capacities. MCI group are at higher risk of suffering from Alzheimer’s disease dementia and other forms of dementia. In the same line, Dodich A. et al. (2016) [21], claimed that ADD patients showed decreased ToM performance due to the presence of basic cognitive dysfunction in these patients. Neurodegenerative dementia patients present simultaneous impairments of different cognitive abilities, and mentalizing deficits may coexist with dysfunctions in other cognitive domains. In the same study, it became obvious that the MCI group did not differ from healthy adults, confirming relevant previous findings [22]. In addition, whereas very few studies have examined this in MCI specifically, ToM abilities, as indexed by the reading the mind in the eyes test (RMET), have been proven to be clearly disturbed in those individuals with amnestic MCI, compared to healthy people [23,24,25]. Several studies have shown that in ADD, those social cognitive deficits display a connection between both volumetric loss and white matter pathology [26,27,28,29]. In the meta-analysis of the ToM–ADD relationships, Bora et al. (2016) [30] presented evidence for an AD-related deficit in ToM; however, it was unclear whether this was caused by problems with both or only one of the subcomponents, as there was no differentiation between the two subcomponents: affective and cognitive ToM. However, there is evidence that in the mild to moderate stages of ADD, key brain regions participating in the overall ToM processing, such as the precuneus, temporal poles, and posterior superior temporal sulcus, are impacted [31]. To sum up, a number of studies have provided evidence for AD-related impairment on one or both ToM components [21,32,33]; others have shown no significant AD-related impairment for either or have reported evidence of a disconnection between the two [34,35].

### 1.3. ToM Abilities and Vascular Pathology

Spiro and Brady (2011) [36] mentioned vascular health as a known predictor of cognitive variability in aging. Vascular risk factors associated with aging, including high blood pressure, can adversely affect cerebral integrity and are linked to lower performance across individual cognitive domains and on measures of global cognitive status [37,38]. Specifically, high blood pressure is associated with reduced learning and memory, slowed processing speed, and reduced executive functions [39,40]. Interestingly, these domains also appear to underlie ToM performance in older age [1]. This functional overlap is an indication that vascular health is a potential, yet unexplored, mechanism for age-related reductions in ToM. More thoroughly, studies found that ToM is affected by aging [41,42,43,44]. A number of the brain regions associated with ToM have also been found to be sensitive to vascular aging. It is the prefrontal regions that mainly seem to be affected, in addition to the parietal and temporal cortices [45,46]. In the same line, metabolic syndrome, as a highly prevalent disease, causes a great deal of issues in individuals and is linked to an increased risk of cardiovascular disease and diabetes mellitus. Metabolic syndrome stimulates inflammatory mediators that disturb the brain metabolism. These mediators can be activated by metabolic inflammation and microvascular disorders and may infer greater damage to the white matter and impair cognitive function [47]. Recent studies claim that metabolic syndrome is a risk factor both for diabetes mellitus and cerebrovascular disorders, and also for the progression of AD [48]. Furthermore, metabolic syndrome can accelerate the onset and progression of cerebral small vessel dysfunctions by altering the structure and function of the blood vessels, which can lead to mild bleeding, white matter damage, and brain atrophy [49]. The majority of metabolic diseases of the brain bring about changes in the neuronal structure and subsequent decline in cognitive functions [50,51,52,53,54,55,56].

Taking the previously mentioned studies into consideration, metabolic diseases of the brain can cause changes in the structure and function of brain areas linked to cognition. In more detail, factors including obesity, hypertension, and hyperglycemia apparently have the greatest effect in reducing cognitive performance and abilities in general [57,58]. Based on the aforementioned literature and the gap that seems to exist in the level of ToM abilities in community-dwelling people with a vascular risk profile, the present study attempted to investigate the performance of a group consisting of older adults diagnosed with vascular risk factors (VRF), as compared to MCI patients and healthy controls, in ToM abilities and especially in sarcasm comprehension.

### 1.4. The Purpose and the Hypotheses of the Study

The purpose of the present study was to investigate the performance differentiation in ToM abilities of older adults with VRF and older adults with MCI and their discrimination from healthy controls. Based on what was reported, the hypotheses were formulated as follows. Based on vascular pathology, older adults with VRF would show lower performance at least in some ToM abilities compared to healthy controls, due to a possible underlying brain pathology (e.g., small vessels disease) (Hypothesis 1a). On the other hand, MCI patients due to ADD would perform at a lower level, compared to healthy controls, at least in some dimensions of ToM, due to possible brain pathology of the ADD type or the combination of AD and vascular pathologies (Hypothesis 1b). No specific hypothesis but the following research question has been formulated as regards the comparison of ToM abilities of older adults with VRF and MCI due to AD pathology: are there any similarities in the ToM abilities of these two groups that could contribute to the thesis that a vascular risk profile could be the underlying ground for the development of an AD pathology?

## 2. Methods

### Participants

The sample comprised 109 adults with Greek origin (24 males, 85 females). Their ages extended from 50 to 85 years (M = 66.09, SD = 9.02). The sample consisted of three groups: (1) community-dwelling older adults who developed VRF (*n* = 41, 9 men and 32 women, Mean Age = 68.06, SD = 7.0); (2) older adults with a diagnosis of MCI due to AD (*n* = 44, 9 men and 35 women, Mean Age = 70.2, SD = 7.0); and (3) healthy controls (*n* = 24, 6 men and 18 women, Mean Age = 54.25, SD = 9.0). The two groups of older adults with VRF and MCI did not differ significantly in age (t(85) = 0.22, *p* > 0.05), education level (*χ*^2^(4,80) = 0.90, *p* > 0.05), and gender (*χ*^2^(2,109) = 0.91, *p* > 0.05). It should be noted that the female gender was overrepresented in all groups. With regard to the third group, healthy adults differed from the other groups in age (t(109) = 52.403, *p* < 0.001) and education level (*χ*^2^(4,1) = 12.071, *p* = 0.017); concisely expressed, healthy controls were younger and more educated.

In relevance to the incorporation criteria for the three groups, neurological examination, neuropsychological assessment, and blood tests have been taken under consideration. Participants in all groups were screened for depressive symptomatology, using the geriatric depression scale-15 [59,60], and persons with scores > 6 were excluded from the study. The ability of simple and complex sentences comprehension was administered with the subscale “auditory perception” from the Boston Diagnosing Aphasia Examination [61]. For the assessment of general cognitive ability and to divide the participants into three groups, MoCA was selected [62,63], standardized with normative data for the Greek population [64].

The group of participants with vascular burden consists of older, community dwelling adults who have reported at least one of the three vascular risk factors (hypertension, hyperlipidemia, diabetes mellitus) during the last year. Participants with VRF were recruited from outpatient clinics from the General Hospital of Katerini, where older adults underwent medical monitoring for the aforementioned symptoms and received the relevant treatments. Reasons for exclusion from the group were (a) diagnosis of MCI or dementia of any type, (b) mental or/and psychiatric disorders, (c) neurological disorder of any type, (d) cancer in the last 5 years, myocardial infarction and cardiac instabilities, (e) complaints of memory difficulties, examined using one question in which the participants could answer about the experience of any type of memory problems with “yes” or “no”. According to the MoCA scores, their general cognitive ability ranged from 25 to 30 (M = 26.7, SD = 1.4). About their educational level, 19 participants had a low educational level (LEL: 0–9 years of education), 9 had a medium one (MEL: 10–12 years education), and 13 were highly educated (HEL: 13 or more years of education).

The MCI group consisted of 15 patients from the neurological outpatient clinic of the General Hospital of Katerini, Greece, and 29 patients of the “Alzheimer Hellas” AD treatment centers in Thessaloniki, Greece. All MCI patients had been diagnosed by neurologists at least six months before and followed relevant medical instructions. The inclusion criterium for the MCI group was diagnosis of minor neurocognitive disorders according to DSM-5, while diagnosis of major neurocognitive disorder and the criteria mentioned for the VRF group were causes of exclusion from this group. It is underlined that for the 29 MCI patients from “Alzheimer Hellas”, diagnosis was supported, moreover, by neuropsychiatric assessment, neuroimaging, such as computed tomography or magnetic resonance imaging, relevant biomarkers, and by consensus of specialized health professionals considered experts in neurocognitive disorders. All MCI patients were evaluated by extensive neuropsychological assessment in order to support the diagnosis, including psychometric assessment for the evaluation of general cognitive abilities, specific cognitive functions, such as verbal fluency and verbal learning, and episodic and everyday memory. It is notable that all MCI patients reported complaints of memory difficulties but they also had specific objective decrements as the neuropsychological examination showed. According to the evaluations of physicians and the neuropsychological examination, the vast majority (over 80%) of MCI patients were of the amnestic type with multiple domains. MoCA scores ranged from 21 to 29 (M = 24.4, SD = 2.1). Importantly, it is underlined that 35 MCI patients had at least one vascular risk factor, which was not a criterion for this group; however, it could not be avoided. Regarding the educational level, 11 participants had a low level, 17 had a medium level, and 16 were highly educated.

At this point, it should be mentioned that even if both of the two aforementioned groups had vascular risk factors, in contrast to the VRF group, MCI participants had a neuropsychological assessment according to which their performance in two episodic memory tests and one test of working memory capacity was significantly lower than the performance of the older adults with a vascular risk profile, which was not found to differ from the performance of healthy controls.

For the control group were selected adults with excellent physical and mental health who do not receive any medication; their hematological check in the last six months did not show any VRF. Due to the difficulty of finding this group, relevant information was given to employees of hospitals, schools and other public services and those who met the criteria were asked to volunteer. Volunteers were asked for a recent hematological check and they could not report subjective problems for their memory capacity. For the reasons above, the third group consisted of younger adults (M = 54.25, SD = 3.7) because no older adults met the aforementioned criteria. Rather as a result of younger age (this is a common condition in Greece), healthy controls had a higher level of education compared to the other groups: only 2 participants had a low educational level, 12 had a medium level, and 10 were highly educated. The MoCA score ranged from 26 to 30 (M = 27.7, SD = 1.33).

In the table below (Table 1) are presented, in detail, all the aforementioned data.

## 3. Instruments

Deficits in reading social cues have been widely reported in neurological and psychiatric disorders. Despite the importance of interpretive abilities, very few tools assess the capacity to read social cues. For the evaluation of “social sensitivity” as the ability to read selected social cues, which can be used to make judgments about the behavior, attitudes and emotions of others, in this study was preferred “The Awareness of Social Inference Test (TASIT)” [65]. TASIT comprises three parts: Part 1—emotion evaluation test; Parts 2–3—test of social inference (minimal and enriched, respectively). In the preset study, we used Part 2 (social inference—minimal test).

According to the authors who developed the tool, the TASIT was designed as a criterion referenced test. Normal participants were expected to perform near ceiling on all subtests. Age and education were not significantly associated with performance. However, in the test of social inference, better educated people performed slightly higher.

### The Social Inference—Minimal Test

Part 2 of the TASIT refers to the circumstances in which the viewer is needed to determine the speaker’s meaning and intentions based upon the dialogue, emotional expressions and other paralinguistic cues while examining a person’s ability to perceive social inference, specifically inferences involved in the use of sarcasm within a minimal context. The viewer receives no external or additional information for making this interpretation. Descriptive for the tool conditions, in the ***Sincere*** exchanges, the targeted speaker means what they are saying. In the Sarcastic exchanges, one of the speakers implies the inverse of what they are saying and intends the recipient to perceive the genuine meaning. Sarcasm exchanges are divided into two different subtypes. In half of the cases, the vignettes are related to Simple Sarcasm, in which participants are being sarcastic but this can be understood mainly by reading the paralinguistic cues. The dialogues in the sincere scenes are identical to the dialogues in the Simple Sarcasm; consequently, if the viewer is unable to distinguish the sarcasm, they will read it as a sincere exchange and misinterpret the intention of the speaker and the meaning. On the other hand, in *Paradoxical Sarcasm* vignettes, the dialogue between the persons does not make sense unless it is understood that one of the participants is being sarcastic. In these scenes, if the case that the viewer does not identify the sarcasm, it is troublesome for them to form sense of the scene, and their decisions with respect to speakers’ incentives and emotions are likely to be inaccurate.

So, in order to discriminate the different exchange types in Part 2, examinees need to draw conclusions about (a) the communicative intentions of the speaker (what they intend to do to their conversational partner), (b) whether they want the literal or non-literal meaning of their message to be believed (what they are saying), (c) their beliefs and knowledge about the situation (what they think), and (d) their emotional state (what they are feeling). What is requested from participants is to watch 15 short-duration scenes (20–60 s) of actors interacting in everyday situations. In each of the scenes, there is a simple exchange between two actors. In a third of the scenes, both actors are being sincere and their thoughts and feelings are congruent with the words that they are using. In another third of the scenes, one of the actors is being sarcastic, but it is only by reading paralinguistic cues that the viewer is able to perceive that the sarcastic actor does not mean what he/she is saying. In the final third of the scenes, the script is paradoxical and can only make sense if one of the actors is being sarcastic. After viewing each scene, the person being tested is instructed to answer 4 questions. The first question concerns what the viewer thinks one of the persons in the scene is “doing”. The second question requires from the viewer what he/she believes someone is trying to *say* to the other person in the vignette. The third question investigates the underlying belief—what the viewer thinks someone is “thinking” in the scenes, and the last question refers to what the viewer thinks someone is “feeling”. The participants have only to respond with saying “yes”, “no” or “don’t know”. Finally, all correct answers were calculated for the 15 scenes of Sincere Sarcasm (5), Simple Sarcasm (5) and Paradoxical Sarcasm (5) comprehension. Hence, the score could range from “0” point to “20 (5 scenes × 4 answers)” points for each type of scene.

## 4. Study Design

The present research is a part of the cross-sectional study of first author’s (Glykeria Tsentidou) doctoral dissertation. At this context, all participants were evaluated with an extensive neuropsychological battery and selected for the purpose of the study, which thoroughly assesses memory capacity, complex executive functioning ability, emotion recognition and social cognition, from which related publications have already emerged [66,67,68].

Following the research planning, for the participants from Katerini, the examination took place in the Daily Care Center of the Dementia Disease Patients unit of the General Hospital of Katerini, while for the participants from Thessaloniki, the examination took place in the two centers of “Alzheimer Hellas”. Evaluations were always done in the morning in a quiet and comfortable office, and the duration of the ToM evaluation was almost 30 min. All participants were examined by the first author.

### Ethics Statement

The participants gave written informed consent at the time of their visit, agreeing that their participation was voluntary and that they could withdraw at any time without giving a reason and without cost. At this point, it must be mentioned that demographic data, such as age, gender, and occupation, were selected. Since these are considered personal data, the European Union law, in existence since 28 May 2018, was applied. According to the law, the use of sensitive personal data is allowed only due to research reasons. Therefore, the participants were informed accordingly, and they also agreed that their personal data could be deleted from the web-database after a written request. The study’s protocol was approved by the Scientific and Bioethics Committee of the Greek Association of Alzheimer Disease and Related Disorders (Scientific Committee Approved Meeting Number: 25/21-06-2016) and followed the principles outlined in the Helsinki Declaration of 1975, as revised in 2008. Moreover, the study has the approval of Hellenic Data Protection Authority, license number: 1971.

## 5. Statistical Analysis

The total number of the variables related to performance in the tests used in the present study was three—sincere, simple sarcasm, paradoxical sarcasm for the total score of correct answers. The data analysis was conducted in SPSS version 25 [69]. The analyses carried out were MANOVA and ANOVA. The main aim of the analysis was to compare the performance of the three groups. Partial eta-squared (η_p_^2^) was used for the estimation of the effect size. The Scheffe post hoc test investigated significant group-level differences. Regarding the differentiation of the three diagnostic groups (MCI patients, older adults with VRF and HC), a MANOVA was performed, using education level and diagnostic group (VRF, MCI, HC) as independent variables, and the three performance-related variables as the dependent ones. At this point, it must be noted that the education level variable was added to the analysis as an independent variable because, according to the TASIT manual, TASIT Part 2 is found to be slightly affected by education (see Method: Instruments section).

## 6. Results

The statistical analysis showed that there was only a main effect of the diagnostic group, F(6,196.00) = 2.599, *p* = 0.019, η^2^ = 0.073. From the subsequent ANOVAs, only the analysis using paradoxical sarcasm comprehension performance as the dependent variable showed that there was a significant effect of the diagnostic group on this variable, F(4,128.890) = 2.638, *p* = 0.038, η^2^ = 0.095 (Table 2).

Subsequently, in Table 3, the mean scores and standard deviations of the performance of the three groups in all evaluated conditions are presented, while in the figure below (Figure 1) are showed the differences as a graph.

As shown in Table 3 and Figure 1, the two pathological groups are as capable of processing sincere exchanges as the healthy controls. This means that they both have the cognitive capacities to process scenes in which ToM is not required, to the extent that it is involved in the comprehension of the other two types of scenes. However, when simple sarcasm comprehension is required, the performance decreases for all groups as a result of the need for ToM recruitment. There is also a tendency of the MCI group’s performance to be lower than that of the other two groups, probably due to their higher difficulty to recruit ToM. Hence, simple sarcasm comprehension seems to be difficult for all participants but without significant differences between their performance. On the other hand, paradoxical sarcasm scenes appear to be easier to comprehend, as regards the ToM abilities required, compared to simple sarcasm scenes. Nevertheless, in this case, there is a clear difference between the performance of pathological groups and healthy controls.

According to the Scheffe post hoc tests, the HC group scored significantly higher than the other two groups: from the VRF group, I-J = 2.89 *p* = 0.007, and from the MCI group, I-J = 2.36 *p* = 0.03. MCI patients and older adults with VRF did not significantly differ in their performance regarding paradoxical sarcasm comprehension. This finding could mean that even if paradoxical sarcasm comprehension does not require high levels of ToM abilities compared to simple sarcasm comprehension, only healthy adults of an advanced age are capable of this type of processing. On the other hand, a vascular risk profile with or without mild cognitive impairment seems to compromise the ToM abilities needed to process paradoxical sarcasm.

Healthy adults and older adults with VRF and MCI did not differ in sincere exchanges comprehension, nor in simple sarcasm comprehension. However, the two pathological groups, that is, older adults with VRF and MCI patients, appeared to display similar comprehension deficits as regards, specifically, paradoxical sarcasm.

## 7. Discussion

The present study examined whether there are differentiated performances on ToM abilities, measured as simple and paradoxical sarcasm comprehension, between two pathological groups of older adults (VRF and MCI) and healthy controls. According to our hypothesis, older adults with VRF were expected to present lower performance, compared to HC, due to a potential underlying brain pathology due to the vascular risk factors (e.g., small vessels disease). MCI patients were also expected to perform at a lower level, compared to HC, due to the AD-type brain pathology or the combination of AD type and vascular pathologies.

According to the statistical analysis, both of the above hypotheses (1a, 1b) regarding the comparison with HC have been confirmed, but only in the case of paradoxical sarcasm comprehension. On the other hand, the two pathological groups’ (older adults with VRF and MCI) performance on the same task was not found to differ.

As regards a possible brain substrate for the deficits of MCI patients in sarcasm comprehension, in many studies on ADD and ToM abilities, social cognitive deficits have been linked to white matter pathology [27,28,29,70,71,72]. The same pathology is also presented in older adults with VRF [73], indicating that metabolic syndrome is associated with brain abnormalities. Parastou Kordestani-Moghadam et al. 2020, [42] support that metabolic syndrome activates inflammatory mediators that disrupt brain metabolism. These mediators can be also activated by microvascular disorders and may further cause damage to the white matter. Taking into account all these findings, it seems that deficits of older adults with VRF and MCI in paradoxical sarcasm comprehension, as compared to healthy adults of advanced age, could be due to the same brain substrate, that is, white matter damage at a higher level, compared to the same type of damage in healthy people. This argument could be enhanced even more by the fact that MCI patients have also a vascular risk profile to a large extent.

Besides white matter damage, it is well known that the neural basis of social cognitive processing is complex, involving a range of cortical and subcortical regions and connective pathways [74]. This network varies depending on task demands, but is broadly thought to include limbic regions (amygdala), the prefrontal cortex and the temporoparietal junction, as well as the anterior cingulate and insular cortex [75,76,77]. The similar ToM deficits of older adults with VRF and MCI, namely, the deficits in paradoxical sarcasm comprehension, could be also due to damage in some of these regions or/and brain networks, mainly as a result of a common vascular pathology.

In any case, the findings that the two pathological groups did not differ from HC in sincere exchanges’ comprehension show that the potential brain damage is still restricted enough to cause severe problems in social cognition but not so restricted as compared to healthy adults of advanced age, who presented low performance only in simple sarcasm comprehension, compared to the other two types of scenes (see Figure 1). It seems that the ToM abilities, which are affected in VRF and MCI people but not in healthy ones, are especially those that require simultaneous processing of linguistic input and paralinguistic input, which do not complement each other in any way, and one must attend to and process both of them separately to understand what happens, as is the case of paradoxical sarcasm.

At the behavioral level, among the neuropsychological decrements of individuals with MCI, mentalizing deficits have been documented and associated with verbal irony comprehension problems [1,22,30,78,79,80]. Mentalizing refers to the capacity to make inferences as regards the mental state of others. Specifically, mentalizing is the ability to meta-represent mental states in order to understand the behavior of oneself or other people [81]. First-order mentalizing is the ability to draw conclusions about a person’s mental states, such as beliefs and thoughts, while second-order mentalizing, a more complex form of ToM, corresponds to the ability to simultaneously adopt two perspectives. Higher metarepresentations are involved in more advanced mentalizing required in the comprehension of ‘indirect speech’ and ‘faux pas’ [82]. In this vein, Achim et al. (2012) [83], showed second-order mentalizing deficits in older adults with MCI, agreeing with Baglio et al. (2012) [22] and Polleti et al. (2013) [71]. Hence, it could be possible that paradoxical sarcasm comprehension requires advanced mentalizing abilities, which MCI patients cannot recruit. Episodic memory and complex executive functioning difficulties could also contribute to the inability of MCI patients to be involved in advanced-order mentalizing [32,81,84,85].

As regards older adults with VRF, who presented a similar performance as MCI patients, there are findings showing that vascular risk, measured as pulse pressure, modified the strength of associations between ToM and age-sensitive cognitive resources, such as processing speed. Specifically, individuals with poor vascular health showed lower performance on ToM tasks [86] compared to their non-ToM reasoning performance, and stronger associations between ToM and verbal memory and processing speed [87]. Hence, it seems that mentalizing abilities, which are more resource demanding compared to non-ToM reasoning, decline in people with VRF due to the underlying vascular pathology, which strengthens the need for verbal memory and processing speed recruitment in order to support mentalizing. In sum, the similarly low performance of MCI patients and community-dwelling older adults with VRF in paradoxical sarcasm comprehension may reflect the same underlying difficulty, that is, advanced mentalizing decline due to developing vascular pathology.

In any case, the findings of the present study are important in terms of revealing that community-dwelling older adults presenting a vascular risk profile but without having a specific diagnosis of cognitive impairment present the same specific deficits in ToM as patients diagnosed with mild cognitive impairment. This may mean that the neuropsychological batteries used to assess cognitive functioning in older adults should include social inference tests in order to find potential specific deficits in social cognition that signal the beginning of cognitive impairment long before the diagnosis of any type of mild neurocognitive disorder.

### 7.1. Limitations

The study has limitations. The use of only one specific ToM test could be a limitation of the study; hence, with more tools administered in evaluation, more differences could be obtained. Initially, as far as the sample is concerned, 29 participants—MCI patients— underwent cognitive enhancement programs, which probably affects the findings because the deficits in the MCI group might have been greater if these 29 patients did not participate in such programs. It must also be noticed that healthy controls were not matched with the other groups in individual–demographic factors, something that should be considered in future studies, if possible, given the composition of the population in Greece. In what has to do with the VRF group, because of the lack of imaging examinations, silent brain infarctions cannot be ruled out (whether cortical or subcortical); this is a possibility that could also apply to the MCI group to some extent and ultimately may affect the performance in the TASIT tasks. Nevertheless, the authors of this study were primarily interested in examining community-dwelling older adults with easy-to-find VRF indices (in terms of access to doctors and examinations), who consider themselves healthy, and comparing them with MCI patients and healthy people in order to identify potential impaired abilities in this sample and argue for the need of intervention programs for this group as well. One more limitation concerns the design and implementation of the study, as the first author, who is the only examiner, was not blinded to the clinical diagnosis of the three groups.

### 7.2. Conclusions

The study aimed to investigate the theory of mind in three different groups: community-dwelling older adults with vascular risk factors, MCI patients, and healthy controls. As shown, older adults with a vascular risk profile present similarly specific deficits in ToM as MCI patients, namely, a lower ability to comprehend paradoxical sarcasm, compared to healthy adults. This finding could be mainly explained by white matter damage and the development of vascular pathology in both groups. In this vein, social inference tests, such as the one examining sarcasm comprehension, could stand out as important tools for “capturing” cognitive impairment in aging from its very first steps.

## Figures and Tables

**Figure 1 brainsci-11-00627-f001:**
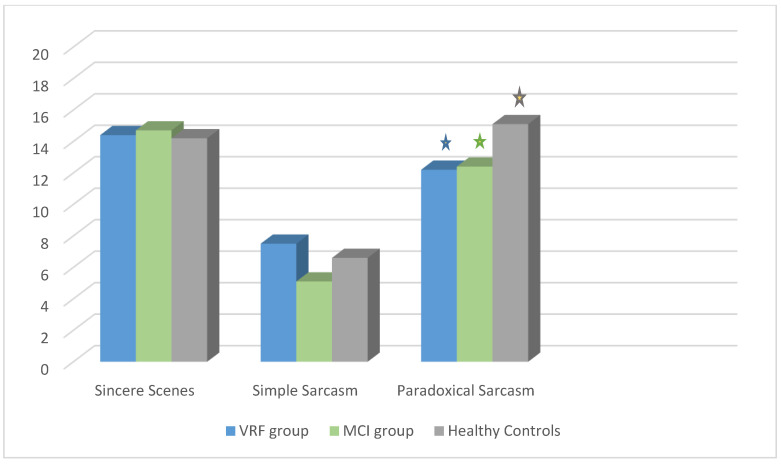
TASIT—Sincere, Simple Sarcasm, and Paradoxical Sarcasm: Mean of adults with vascular risk factors, patients diagnosed with mild cognitive impairment and healthy controls. * Statistical significant results.

**Table 1 brainsci-11-00627-t001:** Demographic data (means and standard deviations of age; frequence of educational levels) and clinical data (means and standard deviations of MoCA scores; and freqence of vascular risk factors) for the three groups.

Demographic/Clinical Data	VRF Group *n* = 41	MCI Group *n* = 44	Healthy Controls *n* = 22
AGE	M = 68.6 SD = 7.0	M = 70.20 SD = 7.0	M= 54.25 SD = 3.7
EDUCATION: 0–9 YEARS	19	11	2
10–12 YEARS	9	17	12
13 AND MORE	13	16	10
MoCA scores	M = 26.7 SD = 1.4	M = 24.4 SD = 2.1	M= 27.7 SD = 1.33
Hypertension	21	18	-
Hyperlipidemia	22	27	-
Diabetes mellitus	11	7	-
Participants with no vascular factors	0	7	22
Participants with 1–2 vascular risk factors	29	33	-
All above vascular risk factors	12	4	-

**Table 2 brainsci-11-00627-t002:** Statistical processing results.

Measures (MANOVA)	Pillai’s V	*p*	η^2^	*p* of Box’s M Test
* Diagnostic Group	0.0146	0.019	0.073	<0.001
Specific Measures (ANOVA)	F	*p*	η^2^	*p* of Levene’s test
Sincere	1.521	0.202	0.057	0.379
Simple Sarcasm	0.961	0.432	0.037	0.031
* Paradoxical Sarcasm	2.638	0.038	0.095	0.607

* Statistically significant results.

**Table 3 brainsci-11-00627-t003:** Mean scores and standard deviations of three groups in TASIT scenes (* *p* < *0*.05).

TASIT Scores	VRF Group M(SD)	MCI Group M(SD)	Healthy Controls M(SD)
Sincere scenes	14.51 (3.6)	14.68 (3.1)	14.79 (3.2)
Simple sarcasm	7.15 (5.4)	5.16 (4.4)	5.96 (4.7)
* Paradoxical sarcasm	11.90 (4.1) *	12.43 (3.4) *	14.79 (3.5) *

* Statistically significant results.

## Data Availability

Not Applicable.

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
