# Peer review of "Similar Theory of Mind Deficits in Community Dwelling Older Adults with Vascular Risk Profile and Patients with Mild Cognitive Impairment: The Case of Paradoxical Sarcasm Comprehension"

_brainsci, 2021, doi:10.3390/brainsci11050627_

Round 1
Reviewer 1 Report
In this work entitled, ‘Similar Theory of Mind deficits in community dwelling older adults with Vascular Risk Profile and patients with Mild Cognitive Impairment: The case of paradoxical sarcasm comprehension’, the authors focused on ToM and in particular on the understanding of sarcasm, which is a more demanding cognitive component than other ToM subdomains as discussed in the paper (especially for paradoxical sarcasms). The sample here studied included 109 older adults, that were distributed into 3 groups (MCI patients, adults with self-reported vascular risk factors [VRF group] and healthy controls) comprising 44, 41 and 24 individuals respectively. Among these 3 groups, and in light of reported literature, the authors explored the two following hypotheses about ToM abilities in older adults with VRF and older adults with MCI and their discrimination from healthy controls:
1/ Hypothesis 1a: Based on vascular pathology, older adults with VRF would show lower performance at least in some ToM abilities compared to healthy controls, due to a possible underlying brain pathology (e.g., small vessels disease).
2/ Hypothesis 1b: MCI patients due to AD would perform at a lower level, compared to healthy controls, at least in some dimensions of ToM, due to possible brain pathology of AD type or the combination of AD and vascular pathologies.
Results showed that the performance of VRF group and MCI patients is not differentiated, while HC performed higher compared to the other two groups. From the authors point of view, these findings may imply that the development of vascular disorder affecting vessels of the brain is associated from its “first steps” to ToM decline at least as regards specific aspects of it such as paradoxical sarcasm understanding.
This result is interesting but appears a bit isolated, with regard to the MocA solely used to determine the overall level of cognition of the enrolled subjects. More detailed and comprehensive neuropsychological assessment would have supported further discussion regarding the afore mentioned hypotheses. Nevertheless, this is how the study was designed, but this should appear in the limitations/weaknesses of the paper. Furthermore, there are sometimes problems of sentence formulation (e.g. the sentence in line 75 to 19 is barely understandable to me).
This study needs several improvements from my point of view, that are indicated below with my remarks and concerns according to each paragraph:
=> Introduction : This part is very long, yet it is didactic. The first 2 paragraphs are very relevant, as is the reference to DSM 5. On the other hand, the following 4 § regarding AD and MCI (lines 71 to 115) are probably too long and could be condensed in one or 2. The same for paragraphs on VRF and metabolic syndrome (lines 116-139) that could be condensed into 1 single section.
=> Methods.
- Participants : a table displaying the different relevant data (i.e. educational level, MoCA scores, GDS scale scores, VRF, etc) would make easier the understanding of the paper. The authors need to specify how the participants were recruited : on local advertisements? On medical decision made by informed physicians? We also need to know if the participants had cognitive complaints or concerns, especially in the VRF and HC groups.
- For the VRF group, the criterion c) neurological disorder any type certainly includes cortical strokes that have been clinically patent (hemianopsia, hemiparesis, aphasic state, etc) but there are actually silent brain infarctions (whether cortical or subcortical) especially in the regions of interest of this study (basal ganglia, dorso-lateral or ventro-median frontal regions). In fact, this must at least be specified in the limitations of the study, or in the discussion (i.e. not only white matter abnormalities but also unnoticed vascular lesions may contribute to the observed differences in ToM abilities , in particular in older MCI or VRF patients compared to HCs).
- For the MCI group (line 198-207): DSM-5 criteria for MCI were used, naturally. But AD patients may have a typical or an atypical cognitive presentation, especially the younger ones (< 65 years). The presence of an amnestic profile with faulty storage is certainly expected in the MCI group, but the number of patients with atypical profile should be specified (possibly inside the table suggested above). Patients harboring logopenic aphasia or Posterior Cortical Atrophy profile (visuo-spatial deficits) may not perform accurately in the tests proposed in the study. This could impact the results of the work and should be added to the methods, and in the discussion if necessary.
- Procedure
The main bias here is that the first author evaluated all participants, and was certainly not blinded to the clinical diagnosis (MCI vs VRF vs HC). This must appear in the limitations section. This is by far the most important bias of the study from my point of view.
Author Response
Manuscript ID: brainsci-1180800
Dear Reviewer,
Thank you for giving us the opportunity to submit a revised draft of the manuscript “Similar Theory of Mind deficits in community dwelling older adults with Vascular Risk Profile and patients with Mild Cognitive Impairment: The case of paradoxical sarcasm comprehension” for publication in the Brain Sciences Journal. We appreciate the time and effort that you dedicated to providing feedback on our manuscript and we are grateful for the insightful comments on and valuable improvements to our paper.
We have incorporated most of the suggestions made by the reviewers. Those changes are highlighted within the manuscript (please see yellow highlights). Follows, a point-by-point response to the reviews.
1) Introduction: This part is very long, yet it is didactic. The first 2 paragraphs are very relevant, as is the reference to DSM 5. On the other hand, the following 4 § regarding AD and MCI (lines 71 to 115) are probably too long and could be condensed in one or 2. The same for paragraphs on VRF and metabolic syndrome (lines 116-139) that could be condensed into 1 single section.
Answer: Thank you very much for this comment! Indeed the Introduction is somewhat long, and after the comment of reviewer 2, it has been enriched.
However, we tried to concoct it, as you suggested it!
2) Participants: a table displaying the different relevant data (i.e. educational level, MoCA scores, GDS scale scores, VRF, etc) would make easier the understanding of the paper. The authors need to specify how the participants were recruited: on local advertisements? On medical decision made by informed physicians? We also need to know if the participants had cognitive complaints or concerns, especially in the VRF and HC groups
Answer: Thank you very much for this reminder! Indeed, it is helpful a table which displaying relevant data. Moreover, missing information had been adding.
3) For the VRF group, the criterion c) neurological disorder any type certainly includes cortical strokes that have been clinically patent (hemianopsia, hemiparesis, aphasic state, etc) but there are actually silent brain infarctions (whether cortical or subcortical) especially in the regions of interest of this study (basal ganglia, dorso-lateral or ventro-median frontal regions). In fact, this must at least be specified in the limitations of the study, or in the discussion (i.e. not only white matter abnormalities but also unnoticed vascular lesions may contribute to the observed differences in ToM abilities , in particular in older MCI or VRF patients compared to HCs)
Answer: Very important clinical observation and thank for that! Indeed the lack of imaging methods in VRF group is a limitation of this study.
4) For the MCI group (line 198-207): DSM-5 criteria for MCI were used, naturally. But AD patients may have a typical or an atypical cognitive presentation, especially the younger ones (< 65 years). The presence of an amnestic profile with faulty storage is certainly expected in the MCI group, but the number of patients with atypical profile should be specified (possibly inside the table suggested above). Patients harboring logopenic aphasia or Posterior Cortical Atrophy profile (visuo-spatial deficits) may not perform accurately in the tests proposed in the study. This could impact the results of the work and should be added to the methods, and in the discussion if necessary
Answer: The majority of MCI patients are monitored in Alzheimer Hellas. Both for the initial diagnosis, and for annual check - ups, specialized physicians (neurologists, psychiatrists, neuropsychologists, and others) evaluated in detail and with all relevant protocols MCI patients, aiming at early diagnosis and the best possible treatment and intervention. In conclusion, MCI patients, at least, participants from Alzheimer Hellas, they are definitely, not misdiagnosed.
5) The main bias here is that the first author evaluated all participants, and was certainly not blinded to the clinical diagnosis (MCI vs VRF vs HC). This must appear in the limitations section. This is by far the most important bias of the study from my point of view.
Answer: Very useful methodological underlining! As mentioned in the manuscript, this article, is a part of doctoral dissertation, so, it is not possible to take this precondition, into the design of this research. Thank you, it is included in the limitations of the study!
We would like to thank the referee again for taking the time to review our manuscript!
Reviewer 2 Report
The manuscript evaluated the Theory of Mind in three different groups of subjects: MCI, elderly subjects with vascular risk factors and healthy controls. The results showed an equal impairment in the comprehension of paradoxical sarcasm in both MCI and elderly with VRF. The results highlighted the importance of VRF on cognitive functioning. Investigation on Theory of Mind are welcome, as it encompasses important brain functions that are not otherwise explored by traditional neuropsychological assessment
That said there are some points that need to be improved
Methods
-Line 170 “MCI due to AD”. Does it mean that patients were positive for amyloid and/or tau biomarkers? Please specify and add reference of the diagnostic criteria (NIA-AA 2018)
-Lines 205-206 “As important, it is noted that, 35 MCI patients had at least one vascular risk factor,” The presence of at least one VRF is the inclusion criteria for the VRF group; please specify the differences between MCI subjects with VRF and subjects with VRF, (MoCa test performances are similar) biomarkers?
- The clinical-demographic features of the groups would be more clear if they are summarized in a Table.
-It is possible to know whether MCI patients were amnestic, non-amnestic or multi-domain MCI, and in which proportion?
-It is possible to specify in the VRF group in which proportion subjects have one or more risk factors? Which was the frequency of each risk factor?
Procedure
-Lines 272-277. This section could be called: Study Design. Its description contains too much information, even if it is not relevant (such as the location of the test administration), and should be rewritten. The reference to previous publications of the first author should be added
Statistical Analysis
-It seems incomplete, most of the information on the statistical analysis is reported i in the Results section.
Results
-Part of this section should be reported in the Statistical part
- I suggest describing the performance results at each ToM test in the different groups, and then the comparison results A table summarizing the single performance of each group would be useful
-Figure1. caption should described and explained the results of the Figure. Moreover statistically significant results should be marked (such as with an * on the top of the corresponding column )
Discussion
-HC were younger and with higher education. Is there information in the literature related to the age and educational effect on ToM performance? Please add a comment.
Line 424 “important tools for “capturing” of cognitive impairment in aging, from its very first steps”, or to evaluate physiological aging? nformation about ToM in elderly are missing. Add also a comment related to the absen
Author Response
Manuscript ID: brainsci-1180800
Dear Reviewer,
Thank you for giving us the opportunity to submit a revised draft of the manuscript “Similar Theory of Mind deficits in community dwelling older adults with Vascular Risk Profile and patients with Mild Cognitive Impairment: The case of paradoxical sarcasm comprehension” for publication in the Brain Sciences Journal. We appreciate the time and effort that you dedicated to providing feedback on our manuscript and we are grateful for the insightful comments on and valuable improvements to our paper.
We have incorporated most of the suggestions made by the reviewers. Those changes are highlighted within the manuscript (please see light blue highlights). Follows, a point-by-point response to the reviews.
- Line 170 “MCI due to AD”. Does it mean that patients were positive for amyloid and/or tau biomarkers? Please specify and add reference of the diagnostic criteria (NIA-AA 2018
4) It is possible to know whether MCI patients were amnestic, non-amnestic or multi- domain MCI, and in which proportion?
Answer: Thank you very much for this comment, indeed some information for MCI patients was missing.
- Lines 205-206 “As important, it is noted that, 35 MCI patients had at least one vascular risk factor,” The presence of at least one VRF is the inclusion criteria for the VRF group; please specify the differences between MCI subjects with VRF and subjects with VRF, (MoCa test performances are similar) biomarkers?
5) It is possible to specify in the VRF group in which proportion subjects have one or more risk factors? Which was the frequency of each risk factor?
Answer: Thank you very much for this comment, indeed some information for VRF group patients was missing.
- The clinical-demographic features of the groups would be more clear if they are summarized in a Table.
Answer: Thank you for this remark! A table has been added.
6) -Lines 272-277. This section could be called: Study Design. Its description contains too much information, even if it is not relevant (such as the location of the test administration), and should be rewritten. The reference to previous publications of the first author should be added
Answer: Thank you for this comment! Of course, previous publications added, and the specific part was modified, although researchers, consider important, to keep all relevant information.
7) It seems incomplete, most of the information on the statistical analysis is reported i in the Results section.
8) Part of this section should be reported in the Statistical part
Answer: Necessary changes have been made, thank you!
9) I suggest describing the performance results at each ToM test in the different groups, and then the comparison results A table summarizing the single performance of each group would be useful
Answer: A table has been added, thank you for this comment!
10) Figure1. caption should described and explained the results of the Figure. Moreover statistically significant results should be marked (such as with an * on the top of the corresponding column )
Answer: Thank you, it has been added!
11) HC were younger and with higher education. Is there information in the literature related to the age and educational effect on ToM performance? Please add a comment.
12) Line 424 “important tools for “capturing” of cognitive impairment in aging, from its very first steps”, or to evaluate physiological aging? nformation about ToM in elderly are missing. Add also a comment related to the absen
Answer: Thank you for this opportunity to include in the Introduction, literature for typical aging in ToM. It is reminded, that performance in TASIT, according to its manual, does not seem to significantly correlate to education, although in the present research, statistically checked the effect of education in the results and no significance pointed.
We would like to thank the referee again for taking the time to review our manuscript!